# Unlocking the Therapeutic Potential of Oral Cancer Stem Cell-Derived Exosomes

**DOI:** 10.3390/biomedicines12081809

**Published:** 2024-08-09

**Authors:** Prabhat Kumar, Rishabh Lakhera, Sadhna Aggarwal, Shilpi Gupta

**Affiliations:** 1Stem Cell and Cancer Research Lab, Amity Institute of Molecular Medicine & Stem Cell Research (AIMMSCR), Amity University Uttar Pradesh, Sector-125, Noida 201313, India; 2Department of Radiation Oncology, University of Texas, MD Anderson Cancer Center, Houston, TX 77030, USA

**Keywords:** oral cancer, cancer stem cells, chemoresistance, extracellular vesicles, exosomes, biomarker, tumor microenvironment

## Abstract

Oral cancer (OC) presents a significant global health burden with rising incidence rates. Despite advancements in diagnosis and treatments, the survival rate for OC patients, particularly those with advanced or recurrent disease, remains low at approximately 20%. This poor prognosis is often due to a small population of cancer stem cells (CSCs) that are capable of self-renewal and immune evasion, playing pivotal roles in proliferation, tumor initiation, progression, metastasis, and therapy resistance. Exosomes, which are nano-sized extracellular vesicles (EVs), have emerged as crucial mediators of cell-to-cell communication within the tumor microenvironment (TME). These vesicles carry diverse molecules such as DNA, RNA, proteins, lipids, and metabolites, influencing various cellular processes. Emerging evidence suggests that CSC-derived EVs significantly promote tumor progression and metastasis and maintain the balance between CSCs and non-CSCs, which is vital for intracellular communication within the TME of oral cancer. Recent reports indicate that oral cancer stem cell-derived EVs (OCSC-EVs) influence stemness, immune evasion, metastasis, angiogenesis, tumor reoccurrence, and drug resistance. Understanding OCSC-EVs could significantly improve oral cancer diagnosis, prognosis, and therapy. In this mini-review, we explore OCSC-derived exosomes in oral cancer, examining their potential as diagnostic and prognostic biomarkers that reflect CSC characteristics, and delve into their therapeutic implications, emphasizing their roles in tumor progression and therapy resistance. However, despite their promising potential, several challenges remain, including the need to standardize isolation and characterization methods and to elucidate exosome-mediated mechanisms. Thus, a comprehensive understanding of OCSC-EVs could pave the way for innovative therapeutic strategies that have the potential to improve clinical outcomes for OC patients.

## 1. Background

Oral cancer (OC) presents a significant global health burden, with rising incidence rates. It predominantly affects the mucosal surfaces of the oral cavity, lips, gums, tongue, and throat [1,2]. It is a significant public health problem in developing countries, including India, where it is the most prevalent cancer among men. It is the sixth most common cancer globally, with India having the highest number of oral cancer cases [3,4]. Despite advances in therapeutic modalities such as surgery, radiation therapy, chemotherapy, and immunotherapy [5], the survival rates for oral cancer, particularly those with advanced or recurrent disease, remain poor and have not substantially improved over the last few decades [6]. Five-year survival remains low (40–50%), mainly because of metastatic invasion, local recurrence, and chemoresistance [7,8,9]. Oral cancer spreads in two primary ways; it invades nearby tissues directly and establishes distant metastases by seeding pre-metastatic niches with secreted elements like exosomes [10]. One of the major hurdles in effectively treating oral cancer lies in its intrinsic heterogeneity and chemoresistance, often attributed to the presence of subpopulations of cells called cancer stem cells (CSCs). CSCs are a small subset of cancer cells with the ability to self-renew and differentiate, contributing to cancer initiation, metastasis, and resistance to conventional therapies [11,12].

Recent studies have increasingly highlighted the significance of molecular pathways and signals mediated by exosomes, which are small extracellular vesicles (EVs) (50–150 nm) released from host cells that facilitate intercellular communication through the transfer of proteins, lipids, and nucleic acids (DNA, mRNA, non-coding RNAs) [13]. These exosomes show great potential in regulating disease progression and metastasis. They are present in various body fluids, including blood, urine, saliva, and cerebrospinal fluid, and in cancer stem cells (CSCs); they can be altered in response to various physiological and pathological conditions, highlighting their potential as biomarkers for cancer diagnostics and therapy. Moreover, exosomes are being explored as therapeutic agents and drug delivery vehicles because of their natural ability to cross biological barriers and deliver their cargo to specific target cells.

Recent research has highlighted the intricate relationship between CSCs and exosomes, revealing that CSC-derived exosomes play crucial roles in modulating the tumor microenvironment, thus promoting angiogenesis, immune evasion, and metastasis [14,15,16,17]. Emerging evidence further suggests that exosomes interact with the TME and may be valuable in diagnosing and treating oral cancer patients. Exosomes derived from oral cancer stem cells (OCSCs) are believed to play crucial roles in creating a tumor-promoting microenvironment, regulating angiogenesis and stemness, mediating drug resistance, and enhancing the metastatic potential of tumor cells [10,18,19]. Understanding the complex interplay between CSCs and exosomes is essential for developing novel therapeutic strategies targeting these elements to inhibit cancer progression and overcome therapeutic resistance.

In this review, we provide an overview of the current understanding of OCSCs and derived OCSC-exos and the isolation and characterization of CSC-derived EVs, highlighting the role of OCSC-EVs in oral cancer development and drug resistance, the challenges involved, and their potential as targets for cancer therapies.

### 1.1. Cancer Stem Cells: Biological Functions and Characteristics

Cancer stem cells (CSCs), also known as cancer stem-like cells, constitute a small, heterogeneous, and highly tumorigenic subset of tumor cells, accounting for approximately 0.05–3% of the tumor population. These cells possess unregulated growth, proliferation, and self-renewal capabilities, allowing them to differentiate into multiple cell types. Such traits are critical for tumor invasion, aggressive metastasis, recurrence, and drug resistance, which are central challenges in cancer pathology and treatment [20,21,22,23]. 

The concept of CSCs dates back to the theories of Virchow and Cohnheim in the 1870s, with definitive identification occurring in 1997 when Bonnet and colleagues isolated a subpopulation in acute myeloid leukemia characterized by specific surface markers (CD34+ but CD38-) [24]. Understanding the biological functions and characteristics of CSCs is essential for developing targeted therapies aimed at eradicating these cells to inhibit cancer progression and overcome therapeutic resistance.

CSCs are particularly noteworthy for their ability to self-renew, a process where CSCs generate new cells that retain stem-like properties crucial for maintaining the CSC population within tumors. These cells can undergo either symmetric division, producing two identical stem-like daughter cells, or asymmetric division, yielding one stem cell and one differentiated non-stem cancer cell [25]. This divisional versatility helps preserve their numbers and contributes to tumor mass expansion, enhancing tumor growth and recurrence potential [26,27]. 

CSCs are characterized by dysregulated signaling pathways, such as Hedgehog, Wnt/β-catenin, PI3K/AKT/mTOR, NOTCH, and JAK-STAT, which play pivotal roles in maintaining their pathological state [28,29,30,31,32,33,34]. These aberrations significantly contribute to CSC deregulation, promoting the epithelial–mesenchymal transition (EMT). EMT is a critical process that facilitates increased stemness, enhanced self-renewal capabilities, and greater invasive and metastatic potential. It also plays a significant role in chemoresistance and the potential for tumor relapse [35,36]. Given the distinctive surface markers and altered signaling pathways exhibited by CSCs, they represent critical targets in developing cancer therapeutics. Targeting these pathways could potentially curb the stemness and associated malignant traits of tumors, thereby offering a promising avenue for enhancing cancer treatment efficacy and patient outcomes.

Techniques such as the dye efflux method have been essential in identifying CSCs by isolating “side population” cells capable of excluding certain dyes. Moreover, the ability of CSCs to form spherical colonies in non-adherent, differential growth factor conditions is another critical hallmark of their identification. Several studies have highlighted the role of putative CSC markers in various cancers, including those of the skin, brain, lung, liver, breast, cervix, prostate, ovary, colorectal region, and head and neck [11,12,37,38,39,40,41]. 

CSCs express distinctive cell surface markers that vary across cancer types; for instance, ALDH+, CD44+, and CD133+ in oral cancer [42]; CD34+ in leukemia [24]; and CD200+ and CD166+ in colorectal cancer [43,44]. Based on these stemness markers, CSCs can be specifically isolated and characterized. These markers are crucial not only for the isolation of CSCs but also for distinguishing them from non-CSC tumor cells. Accumulating evidence shows that CSCs are critically associated with drug resistance, including the upregulation of CD133+ CSCs in glioblastoma xenografts [45]. They promote therapy resistance by expressing multidrug resistance (MDR) transporters, exhibiting more active DNA repair, and inducing stronger apoptotic arrest than other cells, leading to cross-resistance to chemotherapeutic drugs [46,47,48].

Oral cancer stem cells (OCSCs) are critical in tumor initiation, progression, and recurrence. They exhibit self-renewal, differentiation, and resistance to conventional therapies [49]. OCSCs possess specific cell surface markers such as CD44, CD133, and ALDH1, which serve as potential therapeutic targets [12,45]. ALDH1+ OCSCs are notably significant in tumor progression, aggressive metastasis, and therapeutic resistance in oral cancer [11,50,51]. Furthermore, the upregulation of Sox2, Oct4, and Nanog contributes to oral oncogenesis, conferring CSC-like characteristics such as stemness, invasion, and drug resistance [52,53].

Hes1, targeted by activated Notch1, is crucial for maintaining progenitor cell fate [40]. Further, knockdown of Hes1 suppresses the self-renewal capacity of TNFα-treated OSCC cells, suggesting that the Notch1-Hes1 axis is vital for regulating the self-renewal of OCSCs [54]. Recent studies demonstrate that BMI-1, a polycomb group transcription repressor, is essential for maintaining the stemness of OCSCs, including their self-renewal and tumorigenic potential [55,56,57].

A unique characteristic of CSCs is their aggressive metastatic potential. OCSCs drive tumor growth and metastasis through EMT, enhancing their migratory and invasive properties. They are particularly resistant to standard cancer therapies because of their enhanced DNA repair capabilities, quiescent state, and high levels of drug efflux pumps [58,59,60,61,62]. To address these challenges, targeted therapies that specifically eliminate OCSCs without harming normal tissues are imperative. 

Inhibitors targeting pathways regulating OCSC properties [29,33,63,64], such as the METTL3/SALL4 axis, which activates the Wnt/β-catenin pathway post-radiation therapy, are being tested in preclinical trials and offer promising avenues for potentially curative oral cancer therapies [65]. The METTL3/SALL4 axis, which activates the Wnt/β-catenin pathway post-radiation therapy, enhances the CSC phenotype and leads to radio-resistance in oral cancer, representing a potential therapeutic target to eliminate radio-resistant oral cancer cells [66]. Identifying and understanding OCSCs have profound implications for managing oral cancer, and targeting these cells aligns with precision medicine’s goal of tailoring treatments based on individual patient characteristics and tumor genetic profiles. Further research and clinical trials are crucial for translating these findings into effective clinical therapies to improve oral cancer patient outcomes.

### 1.2. Exosomes and Their Cargo

Exosomes have garnered significant attention in recent years because of their critical roles in intercellular communication and potential therapeutic applications [67]. These small, membrane-bound extracellular vesicles (EVs) carry a diverse array of biomolecules, including proteins, lipids, nucleic acids, and metabolites, making them essential components of cellular interactions during numerous physiological and pathological processes [68,69,70]. Initially discovered in the late 1980s as a mechanism for sheep reticulocytes to dispose of transferrin receptors during maturation [71], exosomes have since been recognized for their broader roles in cell-to-cell communication. 

EVs are categorized into three main types based on their origin and size as follows: apoptotic bodies (1000–5000 nm), formed during programmed cell death; microvesicles (200–1000 nm), which bud directly from the plasma membrane; and exosomes (50–150 nm), originating from the endosomal pathway (Figure 1). This classification underscores the diverse nature and roles of EVs in cellular communication.

Exosomes are bilayered vesicles that resemble the plasma membrane of their cell of origin, formed through endocytosis [72]. Key components of exosomes include cholesterol, sphingomyelin, ceramides, receptors, and targeting ligands, which provide structural stability. Proteins such as CD9, CD63, and CD81 (tetraspanins) serve as markers for exosome identification, facilitating their interaction with recipient cells. Heat shock proteins (HSP70, HSP90) in exosomes assist in maintaining protein stability under stress conditions. Proteins involved in exosome biogenesis, such as Alix and Tsg101, and adhesion molecules like integrins (Integrin-α and Integrin-β), play roles in exosome docking and fusion with target cells (Figure 1).

Lipids are crucial for the structural integrity of exosomes, with an enriched composition of cholesterol, sphingomyelin, and ceramides conferring rigidity and stability. These lipids influence the interaction and uptake of exosomes by recipient cells. Nucleic acids, including DNA, mRNA, and non-coding RNAs such as microRNAs (miRNAs), are key components of exosome cargo. These nucleic acids can modulate gene expression and influence cellular behavior in recipient cells. The transfer of miRNAs via exosomes regulates numerous biological processes, including cell proliferation, apoptosis, and immune responses, highlighting their potential as therapeutic delivery vehicles.

Recent studies have elucidated the diverse functions of exosomes in intercellular communication, particularly their roles in cancer progression, metastasis, immune response modulation, and therapeutic delivery [73,74,75,76,77,78]. Exosomes can carry oncogenic proteins and RNAs that promote tumor progression and create a favorable microenvironment for cancer cells. Conversely, exosomes from healthy cells are being explored for their potential to deliver therapeutic agents and modulate immune responses against tumors. The evolving understanding of exosome biology underscores their importance in both physiological and pathological contexts, making them a focal point of current biomedical research.

### 1.3. Exosome Biogenesis

Exosome biogenesis is a complex, tightly regulated process crucial for cell-to-cell communication in various physiological and pathological contexts. The formation of an exosome begins with the invagination (endocytosis) of the plasma membrane to produce an early endosome, which subsequently undergoes further invaginations and sorting of different cargos, leading to the formation of multivesicular bodies (MVBs), also known as late endosomes (Figure 2). Within these MVBs, intraluminal vesicles (ILVs) are formed, containing cargos such as proteins, DNA, RNA, and enzymes. Upon fusion of MVBs with the plasma membrane, ILVs are released as exosomes into the extracellular environment. Alternatively, MVBs can fuse with lysosomes, leading to cargo degradation [78,79]. From here, the MVBs can go through one of two fates, either releasing ILVs into the extracellular space in the form of exosomes, microvesicles, and apoptotic bodies or undergoing degradation after fusing themselves with lysosomes [80].

Recent studies highlight that cargo selection during exosome formation is a selective process driven by proteins and lipids. The ESCRT (Endosomal Sorting Complex Required for Transport) machinery, along with proteins like ALIX and TSG101, plays a key role in sorting ubiquitinated proteins into ILVs [81,82]. Tetraspanins, such as CD63 and CD81, are also crucial for sorting specific bioactive molecules into exosomes, influencing their composition and function. The ESCRT machinery, comprising four protein complexes (ESCRT-0 to ESCRT-III), orchestrates ILV formation and cargo sorting. Proteins destined for ILVs are labeled with ubiquitin and recognized by the ESCRT-0 complex, which recruits ESCRT-I and ESCRT-II for cargo sorting and endosomal membrane budding. ESCRT-III polymerization then facilitates bud scission, releasing ILVs into the MVB lumen (see Figure 2). 

The lipid composition of exosomes, enriched with cholesterol, ceramide, and sphingolipids, dictates membrane curvature and vesicle formation, playing a role in the selective packaging of signaling molecules. Ceramide, produced by neutral sphingomyelinase, is particularly emphasized in exosome biogenesis independent of the ESCRT machinery, suggesting alternative pathways. The functional implications of exosome biogenesis are vast, affecting normal physiological processes such as immune responses and tissue repair, as well as the pathogenesis of diseases, including cancer. Understanding exosome biogenesis has expanded our knowledge of cellular communication and opened new avenues for biomarker discovery and the development of novel therapeutic strategies.

### 1.4. Isolation and Characterization of CSC-Derived Exosomes 

CSC-derived exosomes significantly influence the tumor microenvironment, promote metastasis, and confer drug resistance, making their isolation and characterization vital for developing targeted cancer therapies [80]. Understanding and improving methods for isolating high-purity CSC-derived exosomes are essential for diagnostics and therapeutic development.

CSCs are isolated from tumor tissues or cell lines using flow cytometry or functional sphere formation assays, ensuring a population rich in CSC-specific stemness markers. These cells are cultured in an exosome-depleted medium to promote the exclusive production of CSC-derived exosomes. Once the CSCs reach 70–80% confluency, the medium is collected and subjected to the following series of centrifugation steps: a low-speed centrifugation at 300–500× *g* to remove cellular debris, followed by a higher speed centrifugation at 2000× *g* to eliminate larger vesicles. The supernatant is then ultracentrifuged at 100,000× *g*, resulting in a pellet of crude exosomes [83]. This pellet is resuspended in phosphate-buffered saline (PBS) and further purified through a second ultracentrifugation or through a size-exclusion chromatography (SEC) column (Figure 3). SEC, utilizing materials such as Sephadex or BiogelP, separates exosomes based on size and is particularly effective in achieving high purity, which is essential for downstream analyses [84].

The characterization of exosomes involves morphological and molecular analyses. Electron microscopy, both Transmission and Scanning, is used to examine the size and structure of exosomes, typically ranging from 50 to 150 nm [85,86]. For protein analysis, immunoblotting is employed to detect specific exosomal markers such as CD63, CD9, and TSG101, confirming the exosomal identity and purity. Additionally, ELISA is used to quantify these markers, enhancing the understanding of the surface protein composition of the exosomes. Further nanoparticle tracking analysis (NTA) is used to determine the concentration and size distribution of exosomes, offering quantitative insights that are vital for further biological and clinical evaluations [87,88]. For additional specificity, immunoaffinity capture methods are employed to target exosomal surface proteins (e.g., CD63, CD9) with specific antibodies linked to magnetic beads, allowing for the selective isolation of CSC-derived exosomes. Together, these methodologies not only facilitate the detailed study of CSC-derived exosomes but also enhance our understanding of their biological roles and potential as targets for cancer therapy. 

### 1.5. Role of CSC-Exos in Cancer Development and Drug Resistance

The role of CSC-Exos in cancer development represents a cutting-edge frontier in oncology research, offering novel insights and therapeutic opportunities. The unique ability of CSC-Exos to modulate the tumor microenvironment, influence immune responses, and confer drug resistance distinguishes them from other extracellular vesicles (Figure 4). Unlike bulk cancer cells, CSCs possess a remarkable capacity for self-renewal and differentiation, and their exosomes reflect this unique biology, carrying distinct molecular cargos that drive tumorigenesis and metastasis by modulating the tumor microenvironment [44]. 

Studies indicate that exosomes from diffuse large B-cell lymphoma (DLBCL) cells can induce a CSC phenotype in recipient cells, promoting self-renewal and maintaining stemness through Hedgehog, Wnt, and β-catenin pathways, which provides a new understanding of how CSCs maintain their population and contribute to tumor heterogeneity (Table 1) [89,90].

A notable study demonstrated that mesenchymal stem cell (MSC)-derived exosomes containing miRNA-222 promote quiescence in breast cancer cells, leading to chemo-radiotherapy resistance [111]. Additionally, CSC-Exos are instrumental in the metastatic cascade by preparing pre-metastatic niches, promoting the epithelial–mesenchymal transition (EMT), and increasing matrix metalloproteinase (MMP) expression, which leads to extracellular matrix (ECM) degradation and tumor invasion. The altered expression of miRNA-200 in CSC-Exos further promotes EMT, enhances stemness, and drives tumor invasion and metastasis, contributing to a more aggressive and drug-resistant phenotype. Furthermore, CSC-Exos expressing IL-6, p-STAT3, TGF-β1, and β-catenin promote the generation of cancer-associated fibroblasts and M2 macrophages in colon cancer (Table 1) [112]. 

CSC-Exos are rich in oncogenic factors such as miRNAs, mRNAs, and proteins that enhance proliferation, survival, and resistance to apoptosis in recipient cells. For instance, CSC-Exos enriched with miR-21 and miR-126 have been shown to activate the PI3K/AKT and MAPK/ERK pathways, promoting tumor growth and survival [113]. Additionally, CSC-Exos significantly enhance angiogenesis by transferring pro-angiogenic factors like VEGF and angiopoietin to endothelial cells, promoting blood vessel formation [114]. These exosomes influence the tumor microenvironment by interacting with various stromal cells, including fibroblasts, immune cells, and endothelial cells, promoting the secretion of growth factors, cytokines, and ECM components that create a supportive niche for tumor growth and metastasis. By ensuring a conducive environment for cancer progression, CSC-Exos play a pivotal role in cancer development [92,115].

Moreover, CSC-Exos aid in immune evasion by inducing the polarization of macrophages towards an immunosuppressive M2 phenotype and inhibiting the activation of cytotoxic T cells and NK cells, thereby creating an immunosuppressive microenvironment conducive to tumor growth. For example, CSC-Exos carrying PD-L1 suppress T cell activity and promote immune evasion [116]. Interactions between ovarian cancer stem cells and macrophages through the WNT pathway further promote pro-tumoral and malignant phenotypes in engineered microenvironments [117]. CSC-Exos also transfer drug-resistance factors, including drug efflux transporters, anti-apoptotic proteins, and miRNAs, to sensitive cancer cells, thereby conferring resistance to chemo-radiotherapy and targeted therapies. For instance, CSC-Exos carrying P-glycoprotein (P-gp), a well-known drug efflux transporter, reduce intracellular drug accumulation and efficacy [15,16,17,93,118]. CSC-Exos express PD-L1, leading to the upregulation of PD-1 on CD8+ T cells and promoting their exhaustion. These exosomes contain TGF-β and tenascin-C, which activate mTOR signaling and glycolysis via HIF-1α. Additionally, CSC-Exos carry Notch1, inducing stemness in non-tumor cells and further exacerbating CD8+ T cell exhaustion (Table 1). These findings indicate new therapeutic opportunities for cancer immunotherapy [119]. Understanding these mechanisms of CSC-Exos provides insight into the complex biology of cancer and highlights novel therapeutic targets. Sustained research in this area is crucial for developing effective interventions to counteract the pro-tumorigenic effects of CSC-Exos. Such advancements hold the potential to improve patient outcomes significantly.

### 1.6. Emerging Roles of CSC-Exos in Oral Cancer Progression

The role of CSC-derived exosomes (CSC-Exos) in oral cancer is pivotal in understanding high loco-regional recurrence rates and resistance to chemotherapy. These challenges are primarily due to oral cancer stem cells (OCSCs), which possess unique properties that allow them to evade immune detection and eradication and escape antineoplastic treatment [120]. Recent studies have demonstrated that CSC-derived EVs significantly contribute to tumor progression, drug resistance, metastasis, and angiogenesis, promoting stem-like characteristics in non-CSCs and remodeling the tumor microenvironment (TME) [90,111]. Given their origin, these exosomes carry genetic material from parental cells, suggesting a specific role in oral cancer progression. Hardin et al. (2018) revealed that in the CSC model, exosomes from a CSC clonal line transferred lncRNA MALAT1, SLUG, and SOX2 to normal thyroid cells, but EMT was not induced. However, when these exosomes also transferred linc-ROR, EMT was induced in the thyroid cells, and siRNA targeting linc-ROR reduced invasion. This suggests that CSC-Exos transfer lncRNAs, particularly linc-ROR, to induce EMT and influence the TME and metastatic niche, highlighting potential therapeutic targets [121]. 

Earlier, Mori et al. (2011) showed a positive correlation between M2 TAMs and oral cancer pathological grade in patient specimens [122]. It has also been demonstrated that oral cancer-derived exosomes containing miR-29a-3p promote M2 macrophage polarization. Exosomes from oral cancer cells co-cultured with macrophages increased the expression of M2 markers CD163, CD206, Arg-1, and IL-10. This conditioned medium enhanced oral cancer cell invasion and proliferation, thus promoting tumorigenesis [123]. OC-derived exosomes can reprogram monocytes via the NF-κB pathway and macrophages via miR-29a-3p, thus mediating immunosuppression in the tumor microenvironment. They also carry EMT-promoting cargos such as miR-21 and miR-155, conferring chemoresistance to recipient cells. Additionally, exosomal miR-146a enhances oral cancer stemness, contributing to its resistance to DNA-damaging drugs [124,125,126]. These findings emphasize the importance of immunomodulation by exosomes in developing therapeutic strategies against oral cancer chemoresistance. 

Guiping Zhao et al. (2020) found that exosomes secreted by cancer-associated fibroblasts (CAFs) promote proliferation, migration, and metastasis in esophageal cancer cells by upregulating the Sonic Hedgehog (SHH) signaling pathway, a key regulator of cancer stemness and maintenance [127,128]. Furthermore, inhibiting the exosomal transfer of SHH from CAFs to these cells using cyclopamine suggests a novel therapeutic strategy for treating esophageal carcinoma [127].

Additionally, miR-34a-5p is significantly reduced in CAF-derived exosomes but can be transferred to oral cancer cells, where its overexpression inhibits tumorigenesis by targeting AXL and suppressing proliferation and metastasis via the AKT/GSK-3β/β-catenin pathway. This regulation, which enhances nuclear β-catenin translocation and upregulates SNAIL, drives oral cancer aggressiveness and stemness [129], suggesting the miR-34a-5p/AXL axis as a potential therapeutic target.

A recent study by Patricia Gonzalez-Callejo et al. (2023) identified specific immune cell subsets within EVs derived from HNCs. These EVs selectively targeted MHC-II–macrophages and PD1+ T cells in the HNC-TME [18], serving as therapeutic markers for oral cancer progression and treatment responses [130]. It has been further shown that mesenchymal stem cell-derived exosomes (MSC-Exos) overexpressing miR-126 enhance cell growth, migration, survival, and angiogenesis by targeting the PI3K/Akt and MAPK/ERK signaling pathways [131]. Conversely, miRNA-101 overexpression in MSC-Exos suppressed oral cancer cell proliferation, migration, and invasion. CSC-derived EVs also promote macrophages to exhibit an M2 phenotype. For instance, glioblastoma CSC-generated exosomes (GDEs) preferentially target monocytes, promoting their conversion into immunosuppressive M2 macrophages through upregulated PD-L1 expression due to components of the STAT3 pathway [100].

OCSC-EVs induce a cancer-associated fibroblast phenotype in normal gingival fibroblasts, enhancing the oncogenicity of oral cancer cells. Treatment with ovatodiolide (a bioactive component of *Anisomeles indica*), alone or combined with cisplatin, significantly reduces tumor sphere formation, suppressing stemness and disrupting TME communication, and decreases EV cargos through mTOR, PI3K, STAT3, β-catenin, and miR-21-5p [132]. These results underscore the therapeutic potential of ovatodiolide in treating therapy-resistant OSCCs (Table 2).

A recent study showed that OCSC-derived small EVs from CD133+ CD44+ oral cancer cells transport the lncRNA UCA1, which binds to miR-134, thus modulating the PI3K/AKT pathway through LAMC2. This interaction drives macrophages toward an immunosuppressive M2 phenotype, contributing to tumor growth and inhibiting T cell function. OCSC-derived exosomes polarize tumor-associated macrophages (TAMs) into M2 macrophages by transferring UCA1, which targets the LAMC2-PI3K/AKT signaling pathway, further suppressing anti-tumor immunity including CD4+ T cell activation and interferon-γ production. These mechanisms promote OSCC cell migration, invasion, and tumor growth in xenograft models, facilitated by M2-TAMs influenced by exosomal UCA1 targeting LAMC2 [19]. These insights suggest targeting CSC-derived EVs and M2-TAMs as a potential OSCC therapeutic strategy, highlighting new molecular mechanisms for tumor progression and immunosuppression. Further, OC-CSC-derived EVs specifically interact with M2 macrophages and PD1+ T cells, essential immune constituents in the CSC niche, contributing to an immunosuppression that hinders effective oral cancer therapy [18]. 

Additionally, exosomal TGF-β from oral cancer cells promotes angiogenesis by interacting with epithelial cells and regulating TAM chemotaxis [133]. Oral cancer-derived exosomal thrombospondin 1 (THBS-1) activates M1-like macrophages through p38/Akt/SAPK/JNK signaling, enhancing cancer progression. These M1-like TAMs promote EMT and cancer stem cell formation through the IL-6/Jak/Stat3/THBS-1 axis [134]. These studies underscore the multifaceted roles of CSC-derived exosomes in oral cancer, highlighting their potential as therapeutic targets and biomarkers for monitoring disease progression and treatment responses.

### 1.7. Therapeutic Potential of CSC-Derived Exosomes

CSCs are known for their self-renewal and differentiation abilities and are considered the root cause of tumor initiation, progression, relapse, and resistance to conventional therapies. Exosomes, nanosized vesicles carrying biomolecules, have emerged as key players in cell-to-cell communication within the tumor microenvironment. CSC-derived exosomes, in particular, have gained attention because of their significant therapeutic potential. These exosomes contribute to tumor aggressiveness by reprogramming non-CSCs into stem-like cells, promoting tumor growth and metastasis. They carry molecules like Wnt proteins and non-coding RNAs that enhance chemoresistance and immune evasion, hindering current therapies. However, this characteristic also presents a therapeutic opportunity. By analyzing the unique cargo of CSC-Exos, researchers can identify novel biomarkers for early cancer detection and target vulnerabilities within the CSC population [69,71,78]. For instance, Sánchez et al. (2015) suggested that miRNAs within exosomes derived from prostate cancer stem cells may serve as effective biomarkers for detection and as therapeutic targets [110] (see Table 2).

CSC-derived exosomes can be engineered to deliver therapeutic payloads, including drugs, small RNAs, or immune-modulating agents, directly to cancer cells or the tumor microenvironment. Their ability to bypass biological barriers, target specific cell types, and modulate immune responses makes them promising candidates for novel cancer therapies. For instance, exosomes from breast cancer cell lines have been modified to carry doxorubicin, a chemotherapy drug, reducing cancer proliferation without causing side effects [135]. Targeting CSC-derived exosomes offers a promising therapeutic strategy. Inhibiting the release or uptake of these exosomes could disrupt the supportive communication network within the tumor microenvironment, potentially sensitizing tumors to conventional therapies. For example, CSC-Exos carrying miR-210-3p target the regulation of FGFRL1, potentially acting as a tumor suppressor [17]. Various approaches, including the use of inhibitors of exosome biogenesis and release, have been explored. For example, GW4869, an inhibitor of neutral sphingomyelinase, has shown efficacy in reducing exosome release and sensitizing tumors to chemotherapy [136]. In a study focused on pancreatic cancer, exosomes derived from CSCs were found to be enriched with Glypican-1 (GPC1). These exosomes served as biomarkers, effectively distinguishing between healthy individuals and those with benign or malignant pancreatic tumors [137]. Furthermore, the potential of engineered exosomes loaded with siRNA targeting KRAS in reducing tumor growth in pancreatic cancer models [138] (Table 2).

Furthermore, exosomes can serve as Trojan horses by being loaded with anti-cancer drugs or molecules that disrupt CSC signaling pathways, selectively targeting and eliminating CSCs. Recent research in pancreatic cancer demonstrated the potential of exosomes loaded with siRNA targeting KRAS, resulting in reduced tumor growth in models [138]. CSC-Exos can be engineered to carry certain chemicals that can manipulate the body’s immune response, such as cytokines. When administered, these chemical agents can enhance the efficiency of pre-existing immunotherapies to fight various types of cancers [139,140].

**Table 2 biomedicines-12-01809-t002:** Therapeutic applications of CSC-derived exosomes in cancer therapy.

CSC-Exo Origin	Isolation and Characterization Techniques Used	Therapeutic Potentials of CSC-Exos	Reference
Oral cancer stem cell EVs	Exosome isolation via total exosome isolation reagent and characterization by immunofluorescence imaging	Reduction in miR-21-5p, PI3K, and STAT3, leading to tumor suppression	[132]
Breast cancer stem cell EVs	-	Doxorubicin delivery via CSC-derived exosomes result in a reduction in cell proliferation	[135]
Glioblastoma stem cell EVs	Exosome isolation was achieved by immuno-magnetic based method and characterization by TEM	The overexpression of miR-26a is associated with enhanced tumor characteristics, whereas its inhibition is associated with diminished tumorigenicity	[141]
Lung cancer stem cell EVs	Exosome isolation by the ultracentrifugation method	CSC-derived exosomal miR-210-3p targets FGFRL1, which may have potential as a tumor suppressor	[17]
Liver cancer stem cell EVs	Exosome isolation by the ultracentrifugation method	Exosomes rich in lncRNA-H19 show increased angiogenesis and may be useful as therapeutic targets	[142]
Pancreatic cancer stem cell EVs	Exosome isolation via ultracentrifugation and characterization by TEM and immunogold staining	Surface Glypican-1 may be useful as a biomarker for distinguishing between normal and tumor cells in pancreatic cancer	[137]
Pancreatic cancer stem cell EVs	-	Modified CSC-derived exosomes transfected with siRNA may potentially reduce tumor growth by targeting KRAS	[138]
Prostate cancer stem cell EVs	Isolation via the ExoQuick-TC isolation kit	CSC-derived exosomes containing miR-100-5p and miR-21-5p contribute to prostate cancer progression and may have potential as biomarkers	[110]

### 1.8. Challenges Associated with the Clinical Translation of OCSC-Derived Exosomes

Despite their potential in diagnostics and therapeutics, several significant challenges impede the study and clinical application of OCSC-EVs. These challenges arise from technical limitations and practical issues that need to be addressed for effective utilization.

A primary challenge is the isolation of pure OCSC populations. Many studies rely on stemness markers present on the cell surface to isolate CSCs [12,143,144], but this method is not universally accepted as the most appropriate or standard. The gold standard remains the tumorsphere formation assay in vitro [11,40]. It remains uncertain whether all or most extracted EVs are derived from CSC subgroups, complicating the isolation process. Given the small size of EVs and the presence of a plethora of other EVs, traditional isolation methods often result in the co-isolation of non-specific EVs. This lack of specificity affects purity and yield, hindering downstream analyses. Additionally, harvesting enough EVs from the small percentage of CSCs also poses a significant challenge. 

Once isolated, the comprehensive characterization of OCSC-EVs is equally demanding. While techniques like nanoparticle tracking analysis (NTA), dynamic light scattering (DLS), and electron microscopy (EM) provide initial insights into the size and morphology of EVs, they fall short of providing a complete profile. Advanced molecular profiling, including proteomics, genomics, and lipidomics, is essential to unveil the specific cargo carried by these EVs. However, standardizing these techniques across laboratories remains a significant hurdle because of protocol variations, which lead to inconsistent results.

The functional heterogeneity in OCSC-EVs adds another layer of complexity. These vesicles exhibit diverse biological effects based on their origin and production conditions, hindering the identification of consistent biomarkers and therapeutic targets. Translating OCSC-EV research into the clinic requires addressing regulatory and manufacturing hurdles. Large-scale production, isolation, and characterization protocols must be standardized to ensure safety, efficacy, and reproducibility, which will require rigorous preclinical and clinical testing. Additionally, the safety and stability of exosome-based therapies require thorough evaluation, including understanding the immune response to exogenous exosomes and their potential off-target effects. Advances in nanotechnology and bioengineering are expected to address these challenges, paving the way for clinical applications.

### 1.9. Future Perspective of OCSC-EVs as a Therapeutic Target for Cancer Treatment

The exploration of CSC-derived EVs holds transformative potential for advancing cancer therapy. Forthcoming research aims to uncover their biological properties, molecular contents, and interactions within the tumor microenvironment. Gaining insights into these areas is pivotal for harnessing OCSC-EVs to modulate tumor behavior and intercellular communication, offering new avenues for therapeutic intervention.

OCSC-EVs are particularly promising as targeted drug delivery systems, utilizing their natural tissue-homing abilities to deliver therapeutic agents directly to tumor sites. This capability enhances treatment specificity and reduces systemic toxicity, though challenges remain in developing efficient methods for loading therapeutic molecules into OCSC-EVs and maintaining their stability and functional integrity after delivery.

Additionally, OCSC-EVs hold potential as novel biomarkers for the early detection and monitoring of oral cancer. Their presence in accessible body fluids like saliva and blood makes them ideal for non-invasive diagnostic assays, potentially leading to earlier detection and improved patient outcomes.

The immunomodulatory properties of OCSC-EVs can also be harnessed to boost the effectiveness of existing cancer therapies, including chemotherapy, radiotherapy, and immunotherapy. By modifying the immunological contents of OCSC-EVs, it is possible to stimulate anti-tumor immune responses while minimizing damage to normal tissues, which could significantly improve therapeutic outcomes.

However, the path to the clinical application of OCSC-EVs is fraught with regulatory, manufacturing, and ethical challenges. Standardizing production processes, ensuring safety and efficacy through rigorous clinical trials, and addressing economic and accessibility issues are essential for integrating OCSC-EV therapies into mainstream medical practice.

The future of OCSC-EVs in cancer therapy promises more personalized, effective, and less invasive treatment options. The continued expansion of this field depends on robust, multidisciplinary research efforts aimed at overcoming current barriers and fully exploiting the therapeutic potential of OCSC-EVs. Maintaining momentum in research and development is crucial to significantly alter the treatment paradigm for cancer.

## 2. Concluding Remarks

The multifaceted roles of oral cancer stem cell-derived exosomes (OCSC-Exos) in the progression and resistance of oral cancer underscore their potential as pivotal targets for novel therapeutic strategies. Interfering with the production or uptake of these exosomes, inhibiting their interaction with the tumor microenvironment, or manipulating their molecular cargo opens promising new pathways for the treatment of oral cancer. Furthermore, the utility of OCSC-Exos as biomarkers for the early detection and prognosis of oral cancer offers valuable insights into the molecular dynamics of the disease. Future research dedicated to elucidating these molecular mechanisms and developing therapies based on cancer CSC-EVs could revolutionize the clinical approach to oral cancer, potentially enhancing patient survival rates and quality of life. This forward-looking perspective aligns with the urgent need to bridge laboratory findings with clinical applications, thereby significantly improving outcomes for patients suffering from this deadly disease.

## Figures and Tables

**Figure 1 biomedicines-12-01809-f001:**
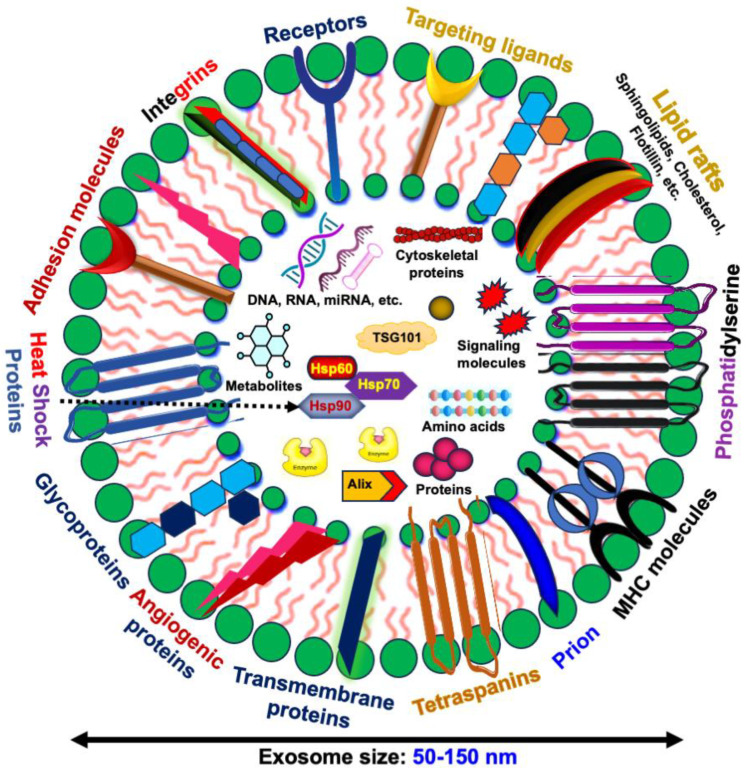
Structure of exosomes and their diverse cargos. These include proteins (tetraspanins, heat shock proteins, enzymes), lipids, nucleic acids (DNA, RNA, miRNA), adhesion molecules, receptors, and signaling molecules.

**Figure 2 biomedicines-12-01809-f002:**
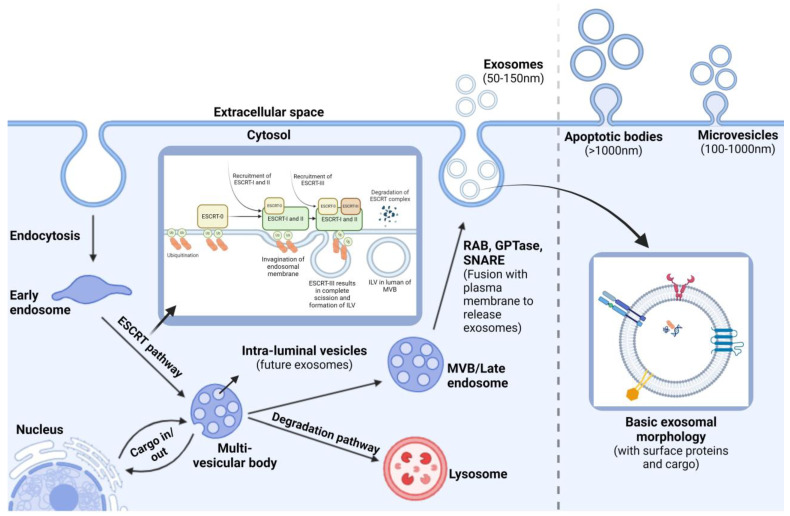
Biogenesis and release of exosomes. Exosomes are small extracellular vesicles enclosed by a lipid bilayer that contains cell surface proteins such as tetraspanins, integrins, flotillins, and transmembrane proteins, which mediate their orientation and interaction with target cells. These vesicles carry a diverse array of biological molecules, including proteins, nucleic acids, lipids, and metabolites, and they modulate the function of recipient cells by delivering their cargos. The biogenesis of exosomes starts with endocytosis, forming early endosomes that mature into late endosomes. During this maturation, intraluminal vesicles (ILVs) are generated within multivesicular bodies (MVBs) through the ESCRT pathway. The fusion of MVBs with the plasma membrane, mediated by Rab GTPase and SNARE proteins, releases ILVs as exosomes into the extracellular space. This figure was created by BioRender (biorender.com).

**Figure 3 biomedicines-12-01809-f003:**
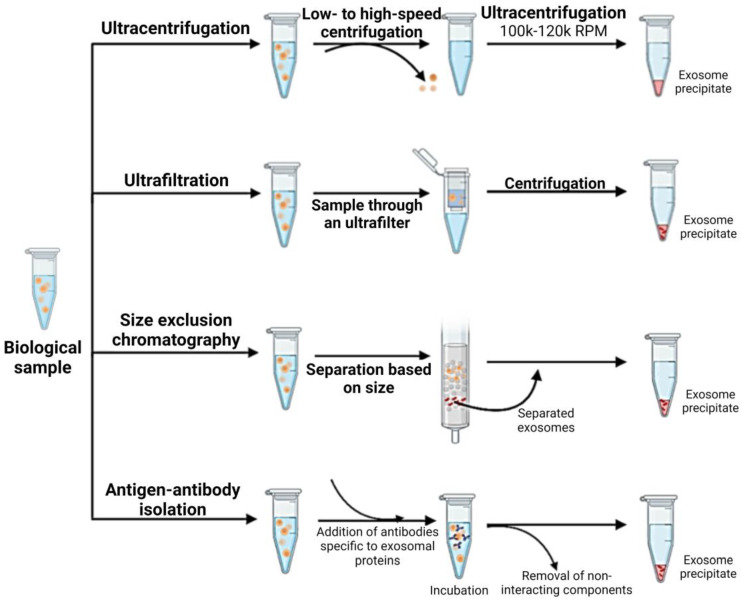
Illustration of various methods for exosome isolation from a biological sample. The techniques include ultracentrifugation, involving low- to high-speed centrifugation steps; ultrafiltration through an ultrafilter; size exclusion chromatography, which separates exosomes based on size; and antigen–antibody isolation, using antibodies specific to exosomal proteins to isolate exosomes selectively, followed by the removal of non-interacting components. This figure was created by BioRender (biorender.com).

**Figure 4 biomedicines-12-01809-f004:**
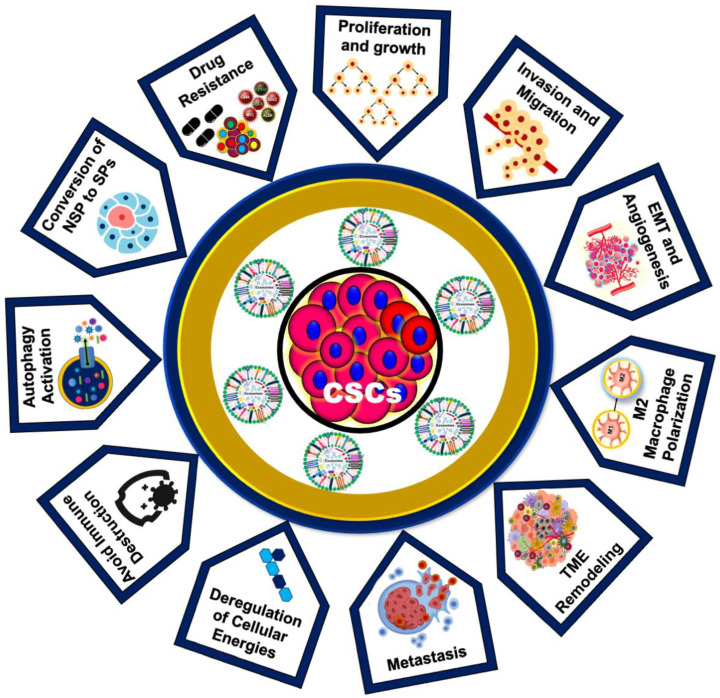
Role of CSC-Exos in cancer progression and drug resistance. The multifaceted roles of CSC-Exos in cancer progression and drug resistance illustrate their contribution to various processes in cancer development.

**Table 1 biomedicines-12-01809-t001:** Functions of CSC-derived exosomes in various cancers.

CSC Source	Cargo	Study Observations	References
Oral cancer stem cells (OC-CSCs) Head and neck cancer stem cells (HNC-CSCs) Esophageal cancer stem cells (EC-CSCs)	miRNAs lncRNA UCA1 O-GlcNAc transferase (OGT) IncRNA FMR1-AS1	○ Reduce tumor growth, increase migration, invasion, and cancer progression. ○ Promote M2 macrophage polarization and suppress CD4+ T cell activity by transferring urothelial carcinoma-associated 1 (UCA1) and targeting the LAMC2-PI3K/AKT pathway. ○ Macrophage-derived exosomal miR- 31-5p promotes tumorigenesis by targeting the LATS2/Hippo signaling pathway. ○ Modulate the tumor microenvironment and the anti-cancer immune response. ○ Overexpression of PD-1 in CD8+ T cells and promotion of immunosuppression. ○ ESC-CSCs exosomes containing FMR1-AS1 may induce stemness in non-CSC cells.	[14,18,19,91,92,93,94,95]
Breast cancer stem cells (BC-CSCs)	miRNAs	○ Target TF-(ONECUT2), resulting in expression of CSC traits. ○ Enhance EMT, metastasis, drug resistance, and survivability in tumor cells.	[96,97]
Glioblastoma stem cells (GB-CSCs)	Tenascin C miRNAs Linc01060 NOTCH1 STAT3	○ Inhibit T cell-induced immune response. ○ Promote proliferation, tumor angiogenesis, invasion, and metastasis. ○ Induce M2 macrophage polarization.	[98,99,100,101]
Lung cancer stem cells (LC-CSCs)	miRNAs	○ Enhances migration capability of cancer cells.	[17]
Gastric cancer stem cells (GC-CSCs)	miRNAs Claudin-7	○ Promote metastasis. ○ CSC-EVs containing cld7 showed increased metastasis.	[16,102]
Colorectal cancer stem cells (CC-CSCs)	miRNAs miRNA-146a-5p Tri-phosphate RNAs	○ Enhance angiogenesis and vascular permeability. ○ Decrease CD8+ T cell infiltration. ○ Sustain MPO+ neutrophil survival associated with cancer progression and poor survival.	[103,104,105]
Pancreatic cancer (PC-CSCs)	miRNAs, CD44v6	○ Enhance chemoresistance and metastasis.	[106,107]
Renal carcinoma (RC-CSCs)	miRNAs, MMPs	○ Promote EMT, angiogenesis, and premetastatic niche formation.	[108,109]
Prostate cancer (PC-CSCs)	miRNAs	○ Contribute to the premetastatic niche and enhance survival of cancer cells.	[110]

## Data Availability

Not applicable.

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
