# Peer review of "Unlocking the Therapeutic Potential of Oral Cancer Stem Cell-Derived Exosomes"

_biomedicines, 2024, doi:10.3390/biomedicines12081809_

Round 1

Reviewer 1 Report

Comments and Suggestions for Authors

In this manuscript, the authors described the therapeutic potential of oral cancer stem cell-derived exosomes in a systematic manner. The potential of CSC-specifically granted diagnostic and prognostic biomarkers were also characterized, offering insights into their therapeutic significance and highlighting their role in tumor progression and treatment resistance. However, so many key points were missing in this manuscript. Therefore, I recommend that these essential descriptions must be included before considering its acceptance for publication.

Minor issues should be addressed.

1.     In the second paragraph of part 1.1, the concept of CSC should perhaps be integrated with the first paragraph. The authors are advised to revise this.

2.     For section 1.1, the paper can be presented by expressing functional and then typological studies, which further connects more smoothly with the section 1.2.

3.     In the manuscript, the author's description of OCSC was too brief and did not seem to correspond to the title. The authors are requested to complement the relevant parts of OCSC-exo. 

Comments on the Quality of English Language

The manuscript should be carefully checked and revised to avoid the spelling, expression and grammar errors. 

Author Response

Manuscript Ref No: biomedicines-3136896

Point-wise-responses to the Reviewer’s Comments on our manuscript entitled “Unlocking the therapeutic potential of oral cancer stem cell-derived exosomes”

Reviewer 1: Minor issues should be addressed.

We express our gratitude to the editor and both reviewers for careful evaluation of our manuscript and their constructive feedback and suggestions helped us to improve it substantially. A point-wise response to both the Reviewer’s comments and respective modifications in the revised manuscript are described below.

Q1. In the second paragraph of part 1.1, the concept of CSC should perhaps be integrated with the first paragraph. The authors are advised to revise this.

Response 1: Thanks to referee for this suggestion. As suggested, we have integrated ‘the concept of CSC…’ with the first paragraph in the revised manuscript.

Q2. For section 1.1, the paper can be presented by expressing functional and then typological studies, which further connects more smoothly with the section 1.2.

Response 2: Thank you for your helpful suggestion. As suggested, we have combined 1.1 and 1.2 sections and frame as one in revised manuscript (highlighted in green colour in page no. 2-4).

Q3.  In the manuscript, the author's description of OCSC was too brief and did not seem to correspond to the title. The authors are requested to complement the relevant parts of OCSC-exo.

Response 3: Thanks to the referee for this suggestion. We have carefully reviewed the literature and have included two additional studies on OCSC-Exos that were not previously cited in our manuscript. We have also tried to enhance and improve this section to provide a more comprehensive description of OCSC and OCSC-exo as suggested.

Q4. Comments on the Quality of English Language: The manuscript should be carefully checked and revised to avoid the spelling, expression and grammar errors.

Response 4: We thank the reviewer for the helpful comments. We have thoroughly revised the manuscript to correct spelling, expression, and grammatical errors, ensuring overall improvement in the quality of the language.

Reviewer 2 Report

Comments and Suggestions for Authors

The article provides a compelling overview of the mini-review and its focus on OCSC-derived exosomes in oral cancer. With some minor refinements, it can be even more impactful and will be suitable for publication.

1.     Consider adding a sentence or two about the structure of the mini-review. For example, briefly mention the main sections or key themes that will be covered. This would provide readers with a clearer roadmap of the content.

2.     Section 1.1 and 1.2 is merely the repetition of the text with similar wordings. Add more information in section 1.2 other than role of OCSCs, EMT, markers and pathways or combine both the section to frame as one.

3.     Provide references corresponding to the isolations of CSC EVs.

4.     Line “Heather Hardin et al (2018) revealed that in the CSC model, exosomes from a CSC clonal line transferred lncRNA MALAT1, SLUG, and SOX2 to normal thyroid cells, but EMT was not induced” should be replaced with “further, Hardin et al (2018) revealed…. but EMT was not induced”.

5.     Provide a bit more detail on the specific challenges associated with the study and use of OCSC-EVs, particularly in the context of isolation and characterization. This would enhance the reader’s understanding of the hurdles that need to be overcome.

6.     Provide a table describing the recent studies related to therapeutic potential of CSC-derived exosomes for cancer treatment.

7.     Provide a section of future perspective of OCSC-EVs as therapeutic solution to cancer.

Comments on the Quality of English Language

Minor editing of English is required

Author Response

Manuscript Ref No: biomedicines-3136896

Point-wise-responses to the Reviewer’s Comments on our manuscript entitled “Unlocking the therapeutic potential of oral cancer stem cell-derived exosomes”

Reviewer 2: Comments and Suggestions for Authors: The article provides a compelling overview of the mini-review and its focus on OCSC-derived exosomes in oral cancer. With some minor refinements, it can be even more impactful and will be suitable for publication.

Reviewer 2: We thank the reviewer for the encouraging comments and constructive suggestions. Based on the reviewer's feedback, we have made the following refinements to enhance the impact and clarity of our manuscript.

Q1. Consider adding a sentence or two about the structure of the mini-review. For example, briefly mention the main sections or key themes that will be covered. This would provide readers with a clearer roadmap of the content.

Response 1: We thank the reviewer for the helpful comments. In response to the suggestion, we have added a brief outline of the structure of the mini-review to the introduction section (page no. 2).

Q2. Section 1.1 and 1.2 is merely the repetition of the text with similar wordings. Add more information in section 1.2 other than role of OCSCs, EMT, markers and pathways or combine both the section to frame as one.

Response 2: Thank you for your helpful suggestion. We have carefully reviewed sections 1.1 and 1.2 and found overlapping content. As suggested, we have combined sections 1.1 and 1.2 into a single section to avoid redundancy and provide a more cohesive narrative. The revised section now includes additional information beyond the role of OCSCs, EMT, markers, and pathways. These changes are highlighted in green on pages 2-4 of the revised manuscript.

Q3. Provide references corresponding to the isolations of CSC EVs.

Response 3: Thank you for your valuable suggestion. We have included references that specifically address the isolation of CSC-EVs.

Q4. Line “Heather Hardin et al (2018) revealed that in the CSC model, exosomes from a CSC clonal line transferred lncRNAMALAT1, SLUG, and SOX2 to normal thyroid cells, but EMT was not induced” should be replaced with “further, Hardin et al (2018) revealed…. but EMT was not induced”.

Response 4: Thank you for the suggestion. As suggested, the references have been reformatted accordingly.

Q5. Provide a bit more detail on the specific challenges associated with the study and use of OCSC-EVs, particularly in the context of isolation and characterization. This would enhance the reader’s understanding of the hurdles that need to be overcome.

Response 5: As suggested, we have added a new section on the specific challenges associated with the study and use of OCSC-EVs in the revised manuscript (section 1.8, page no. 16-17).

Q6. Provide a table describing the recent studies related to therapeutic potential of CSC-derived exosomes for cancer treatment.

Response 6: Thank you for the suggestion. In response to the reviewer's comment, we have included Table 2 (see Table 2 on page no. 15-16) in the revised manuscript that describes relevant studies related to the therapeutic potential of CSC-derived exosomes for cancer treatment.

Q7. Provide a section of future perspective of OCSC-EVs as therapeutic solution to cancer.

Response: Thank you for your insightful suggestion. As suggested, we have now added a section (section 1.9) of future perspective of OCSC-EVs as therapeutic solution to cancer in the revised manuscript on page no. 17.

Q8. Comments on the Quality of English Language: Minor editing of English is required

Response 8: We thank the reviewer for the helpful comment. We have thoroughly revised the manuscript to correct minor grammatical errors and improve the overall quality of the language.

We hope the referees and the editor will find our point-wise responses to their quarries satisfactory and the revised manuscript will now be acceptable for publication in your esteemed journal, Biomedicines. 
